# Targeted Modulation of Chicken Genes In Vitro Using CRISPRa and CRISPRi Toolkit

**DOI:** 10.3390/genes14040906

**Published:** 2023-04-13

**Authors:** Brittany Chapman, Jeong Hoon Han, Hong Jo Lee, Isabella Ruud, Tae Hyun Kim

**Affiliations:** 1Department of Animal Science, The Pennsylvania State University, University Park, PA 16802, USA; 2The Huck Institutes of the Life Sciences, The Pennsylvania State University, University Park, PA 16802, USA

**Keywords:** activation, chicken, CRISPR, dCas9, interference

## Abstract

Engineering of clustered regularly interspaced short palindromic repeats (CRISPR) and the CRISPR-associated protein 9 (Cas9) system has enabled versatile applications of CRISPR beyond targeted DNA cleavage. Combination of nuclease-deactivated Cas9 (dCas9) and transcriptional effector domains allows activation (CRISPRa) or repression (CRISPRi) of target loci. To demonstrate the effectiveness of the CRISPR-mediated transcriptional regulation in chickens, three CRISPRa (VP64, VPR, and p300) and three CRISPRi (dCas9, dCas9-KRAB, and dCas9-KRAB-MeCP2) systems were tested in chicken DF-1 cells. By introducing guide RNAs (gRNAs) targeting near the transcription start site (TSS) of each gene in CRISPRa and CRISPRi effector domain-expressing chicken DF-1 cell lines, significant gene upregulation was induced in dCas9-VPR and dCas9-VP64 cells, while significant downregulation was observed with dCas9 and dCas9-KRAB. We further investigated the effect of gRNA positions across TSS and discovered that the location of gRNA is an important factor for targeted gene regulation. RNA sequencing analysis of *IRF7* CRISPRa and CRISPRi- DF-1 cells revealed the specificity of CRISPRa and CRISPRi-based targeted transcriptional regulation with minimal off-target effects. These findings suggest that the CRISPRa and CRISPRi toolkits are an effective and adaptable platform for studying the chicken genome by targeted transcriptional modulation.

## 1. Introduction

Clustered regularly interspaced short palindromic repeats (CRISPR)/CRISPR-associated protein 9 (Cas9), which was first discovered in the bacterial immunity system [1], is a powerful tool for genome editing in living organisms [2,3]. This technology requires two components for DNA double-strand breaks (DSBs) on the target locus: a guide RNA (gRNA) and the Cas9 nuclease that induces the cleavage by binding a gRNA to complementary regions in the genome and subsequent endonuclease activity.

Not only was Cas9 designed for gene knockout by inducing DSBs, but it was repurposed to allow for gene regulation rather than genome editing by utilizing catalytically inactive Cas9 lacking endonuclease activity (dCas9) and fusing effector proteins [4,5,6,7,8,9]. The CRISPR/dCas9-mediated gene activation (CRISPRa) system enhances endogenous gene expression with dCas9-mediated recruitment of activator domains by targeting the transcription start site (TSS) or enhancers [10,11]. The fusion of dCas9 and activator domains such as VP64 [12], VPR (containing VP64, Nuclear Factor p65, and Epstein–Barr virus R transactivator (Rta) domains) [10,13], p300 histone acetyltransferase [14,15], SunTag and the synergistic activation mediator (SAM) system allowed for the activation of the expression of coding and noncoding target loci ablation of DNA sequences [16,17]. The CRISPR/dCas9-mediated gene interference (CRISPRi) system can utilize gRNAs for repression of singular or multiple genes by blocking transcription elongation or initiation [18]. Transcription repression via the CRISPRi system is reversible, as the dCas9-repression domain blocks transcription of the gene, rather than editing the genetic information [4]. A construct with the fusion of the dCas9 protein and repressor domains such as Krüppel-associated box (KRAB) and the methyl-CpG binding protein 2 (MeCP2) reduced gene expression by recruiting corepressors or transcriptional regulators resulting in epigenome modification [16,19,20,21].

Due to their versatility and simplicity, the dCas9 systems, CRISPRa and CRISPRi, are widely used to regulate endogenous gene expression in a variety of species through in vitro and in vivo experiments [16,18,19,20]. Zebrafish and mice are often used for in vivo experiments and human experiments are mostly conducted in vitro [22,23]. However, the methods of delivery for some of these experiments are moving from in vitro to in vivo [24,25]. In chicken, CRISPRa using SAM and CRISPRi using KRAB or lysine-specific histone demethylase 1 (LSD1) domains were also applied to induce gene activation or repression in chicken embryos [26]. The CRISPRa system using the VPR activator domain system was adapted to elevate the gene expression of one of the major chicken egg white proteins, ovalbumin, in chicken DF-1 and embryonic fibroblasts [27]. Moreover, the dCas9-VP64 was used to activate viral genes in chicken lymphoblastoid cell lines (LCLs) [28], suggesting its versatility in chicken gene regulation.

CRISPR/Cas9 technology in livestock species, including poultry, is continuously expanding. Through the use of these genome-editing technologies, the possibility of producing fitter, healthier, and more productive animals continues to grow [29,30,31,32]. It has been reported that the use of CRISPR/Cas9 systems in chickens has assisted in increasing productivity, disease resistance, egg composition, and can also have an impact on animal welfare concerns [33].

The limited use of the CRISPR-based toolkit in chickens warrants a systemic evaluation of the CRISPRa and CRISPRi systems with diverse effector domains and different gRNA locations for enabling effective transcriptional regulation in chickens in order to make better use of this powerful platform in avian research. Therefore, in this study, the effects of the three different types of CRISPRa (VP64, VPR, and p300) and CRISPRi (dCas9, dCas9-KRAB, and dCas9-KRAB-MeCP2) systems in chicken DF-1 cells were evaluated and compared. The effective transcriptional regulation effect was evaluated at the transcriptome level in the CRISPRa and CRISPRi DF-1 cell lines by bulk RNA sequencing.

## 2. Materials and Methods

### 2.1. Culture of Chicken DF-1 Cells

Chicken DF-1 cells [CRL-12203; American Type Culture Collection (ATCC), Manassas, VA, USA] were maintained and sub-passaged in Dulbecco’s minimum essential medium (DMEM; Hyclone, Logan, UT, USA) supplemented with 1× antibiotic-antimycotic (ABAM; Thermo Fisher Scientific, Waltham, MA, USA) and 10% fetal bovine serum (FBS; Hyclone). DF-1 cells were incubated at 37 °C with an atmosphere of 5% CO_2_ and 60–70% relative humidity.

### 2.2. Generation of CRISPRa Cell Line

To generate the CRISPRa cell line, the CRISPR/Cas9 and nonhomologous end joining (NHEJ) pathway (CRISPR/Cas9-NHEJ)-mediated genome editing strategy was applied [34]. First, the CRISPR/Cas9 plasmids, pSpCas9 2A-Puro (PX459) (a gift from Feng Zhang, Addgene plasmid #62988) targeting the 3′ region of chicken glyceraldehyde 3-phosphate dehydrogenase (*GAPDH*) gene (GAPDH #1) and targeting the common region of the three CRISPRa vectors (CRISPRa #1), Cas9m4-VP64 (Addgene plasmid #47316) [35], SP-dCas9-VPR (Addgene plasmid #63798) [13], or pcDNA-dCas9-p300 Core (Addgene plasmid #61357) were constructed [35,36]. The 2.5 × 10^5^ DF-1 cells were then transfected with 1 µg of GAPDH #1, 1 µg of CRISPRa #1, and 2 µg of each CRISPRa vector using Lipofectamine 3000 according to the manufacturer’s instructions. The transfected cells were treated with Geneticin Selective Antibiotic (G418, 300 µg/mL) (Thermo Fisher Scientific, Waltham, MA, USA, 24–48 h post transfection, and the drug selection was maintained to establish cell lines for at least 2 weeks. The gRNA and oligo sequences used in all-in-one CRISPR/Cas9 vector construction are listed in Appendix A.

### 2.3. Generation of CRISPRi Cell Line

To generate the CRISPRi cell line, 2.5 × 10^5^ DF-1 cells were transfected with 200 ng of *piggyBac* transposon vectors containing CRISPRi components (pB-CAGGS-dCas9, pB-CAGGS-dCas9-KRAB or pB-CAGGS-dCas9-KRAB-MeCP2) which were gifts from Alejandro Chavez and George Church (Addgene plasmid #110823, #110822, and #110824, respectively) and 50 ng of transposase vector (PB200, Systems bioscience, Palo Alto, CA, USA) using the Lipofectamine 3000 (Thermo Fisher–Scientific) according to the manufacturer’s instructions [21]. The transfected cells were treated with blasticidin (3 µg/mL) (Thermo Fisher–Scientific) 24–48 h post transfection, and the drug selection was maintained to establish cell lines for at least 2 weeks.

### 2.4. Validation of CRISPRa and CRISPRi Cell Line

To confirm the expression of CRISPRa and CRISPRi components in the cell lines, total RNA from each cell line was isolated using Direct-Zol RNA Mini or Microprep kit (ZymoResearch, Irvine, CA, USA) and reverse-transcribed using High-Capacity cDNA Reverse Transcription Kit (Thermo Fisher–Scientific) [37]. Then, the cDNAs of each cell line were amplified with the CRISPRi and CRISPRa component-specific primers by polymerase chain reaction (RT-PCR). All reactions were performed under the same conditions using DreamTaq Green DNA Polymerase (Thermo Fisher–Scientific) according to the manufacturer’s instructions. The cycling conditions were as follows: 95 °C for 1 min, 34 cycles of 95 °C for 30 s, 60 °C for 30 s, 72 °C for 1 min, followed by a melting cycle. The primers used in the PCR were listed in Appendix A.

To validate the targeted gene insertion of CRISPRa vectors, the genomic DNA of the cell lines was extracted by using Quick Extract DNA Extraction Solution (Lucigen, Middleton, WI, USA) according to the manufacturer’s instructions. Then, the genomic DNAs were analyzed by PCR using knock-in-specific primers (Appendix A). The amplicons were cloned into the pGEM-T-easy vector (Promega, Madison, WI, USA) and sequenced using T7 primer. The sequences were then compared against reference sequences using SnapGene (GSL Biotech, LLC, San Diego, CA, USA).

### 2.5. gRNA Expressing Vector, gRNA Design and Transfection

For transient expression of gRNAs, the vector containing guide RNA (gRNA) scaffold driven by human U6 promoter and puromycin resistant gene was synthesized (Genewiz, South Plainfield, NJ, USA). gRNAs targeting each gene or mock controls were then cloned into the synthesized vector by *BbsI* restriction enzyme (New England Biolabs, Ipswich, MA, USA) digestion and subsequent ligation [36]. Five gRNAs for interferon regulatory factor 7 (*IRF7*), and three gRNAs each for peroxisome proliferator-activated receptor γ (*PPARG*), high mobility group AT-hook 1 (*HMGA1*) and SWI/SNF related, matrix associated, actin dependent regulator of chromatin, subfamily b, member 1 (*SMARCB1*) were designed using the CHOPCHOP algorithm (https://chopchop.cbu.uib.no/, accessed on 14 February 2022) [38]. For mock control, three gRNAs were designed that were not complementary to the chicken genome. TATA box for each gene were predicted by TFBIND (https://tfbind.hgc.jp/, accessed on 14 February 2022) [39]. The oligos used in gRNA vector construction are listed in Appendix A. An amount of 2 µg of the constructed individual gRNA vectors was then transfected into the established CRISPRi and CRISPRa cell lines using Lipofectamine 3000 (Thermo Fisher–Scientific) according to the manufacturer’s protocol. For the combination gRNA for *SMARCB1*, 1.5 µg of gRNAs was used—500 ng of each gRNA (gRNA1, 2, and 3). For *IRF7* combination gRNA, 2 µg of gRNAs was used—400 ng of each gRNA (gRNA1, 2, 3, 4, and 5). The transfected cells were treated with puromycin (1 µg/mL) (Thermo Fisher–Scientific) and harvested 48–72 h after transfection, followed by RNA extraction using the Direct-Zol RNA Mini or Microprep kit (ZymoResearch). Off-targets of each gRNA were determined using CRISPOR (http://crispor.tefor.net/, accessed 29 July 2022) [40]. The off-targets were screened up to three mismatches of gRNA.

### 2.6. Quantitative Real-Time Polymerase Chain Reaction (qRT-PCR)

Total RNA was used to synthesize cDNA by using High-Capacity cDNA Reverse Transcription Kit (Thermo Fisher–Scientific). For qPCR, the PowerUp SYBR Green Master Mix and protocol (Thermo Fisher–Scientific) was used. For each reaction, 2 µL of cDNA and 1 µL of each forward and reverse primers were used for a 20 µL qPCR reaction. Cycling conditions were as follows: 50 °C for 2 min, 95 °C for 2 min, 40 cycles of 95 °C for 15 s and 60 °C for 1 min, followed by a melting cycle. Each gene expression level was normalized to the housekeeping gene *GAPDH* expression using the ΔΔCt method [41]. All qPCR was performed using at least 3 biological replicates, and a significant difference compared to mock control was evaluated by *t*-test.

### 2.7. Bulk RNA Sequencing and Data Analysis

A total of 8 cDNA libraries (2 replicates each) were prepared with NEBNext Ultra™ II RNA Library Prep Kit (New England Biolabs) according to the manufacturer’s protocol. Each library was sequenced at a minimum of 20 million, 150 bp paired-end reads per sample. The FASTQ read files from RNA-seq analysis have been deposited in NCBI’s Gene Expression Omnibus (GSE217310). We checked the quality of each FASTQ read file using fastQC (version 0.11.9) and trimmed the adaptor sequence with TrimGalore (version 0.6.7). The trimmed FASTQ files were aligned against bGalGal1.mat.broiler.GRCg7b (NCBI annotation release 106) reference chicken genome using STAR aligner (version 2.7.10a) [42]. Raw read counts were extracted by HTSeq (version 0.13.5) from each aligned bam file and used to identify differentially expressed genes (DEGs) [43]. EdgeR R package (version 3.38.4) was used to identify DEGs between transcriptomes (false discovery rate (FDR) < 5%) [37]. Functional annotations for significantly differentially expressed genes were performed using DAVID 2021 [44,45]. The enriched gene ontology (GO) terms on biological processes and the pathways obtained from DAVID functional analysis were filtered for significance by gene count ≥ 5 and *p*-value < 0.05.

## 3. Results

### 3.1. Establishment of CRISPRa and CRISPRi Chicken DF-1 Cell Lines

To examine the effectiveness of diverse CRISPRa and CRISPRi systems in chickens, we established multiple CRISPRa and CRISPRi DF-1 cell lines. To establish the CRISPRa DF-1 cell lines, CRISPR/Cas9-nonhomologous end joining (NHEJ) mediated gene targeting methods were adapted [34]. To induce constant and robust expression of the inserted gene, the 3′ region of the housekeeping gene *GAPDH* was targeted without disturbing gene expression (Appendix A). The targeted cleavage of all-in-one CRISPR vector targeting the 3′ region of the *GAPDH* gene (GAPDH#1) was validated by T7E1 assay and subsequent sequencing analysis (Appendix A). The all-in-one CRISPR vector GAPDH #1 was then transfected to chicken DF-1 cells with the donor plasmids containing CRISPRa effectors (dCas9-VP64, dCas9-VPR, or dCas9-p300) (Figure 1A) and another all-in-one CRISPR vector targeting the CRISPRa donor plasmids (CRISPRa #1) for linearization of the donor. After the antibiotic selection using G418, successful integration of the donor plasmid into the targeted region was confirmed by genomic DNA PCR and sequencing of the PCR products (Appendix A). The RT-PCR results showed that the CRISPRa components, VP64, p65 or p300, were strongly expressed in each cell line (Figure 1B). These results confirmed the successful establishment of the CRISPRa DF1 cell lines. For CRISPRi cell line establishment, *piggyBac* transposition allowing direct integration of the gene of interest into the genome was adapted [46]. The *piggyBac* vectors containing CRISPRi effectors (dCas9, dCas9-KRAB or dCas9-MeCP2) (Figure 1C) were transfected to chicken DF-1 cells with *piggyBac* transposase and stably integrated cells were selected with blasticidin for at least 2 weeks. After the antibiotic selection, RT-PCR was performed and the expression of dCas9, KRAB or MeCP2, were successfully detected in the established CRISPRi DF-1 cell lines (Figure 1D).

### 3.2. CRISPRa and CRISPRi-Mediated HMGA1, SMARCB1, IRF7 and PPARG Regulation

To demonstrate the transcriptional regulation of CRISPRa and CRISPRi in chicken cell lines, four genes with diverse expression levels in DF-1, *HMGA1, SMARCB1, IRF7* and *PPARG* were selected that have FPKM values of 107, 56, 4.5 and 1.2, respectively. Three (*HMGA1*, *SMARCB1*, and *PPARG*) or five (for *IRF7*) gRNAs targeting near the TSS of each gene were designed and the gRNA expression vector was constructed. For *HMGA1*, the three gRNAs were designed in locations within −161 bp to +40 bp (1, 2, and 3), and all of the gRNAs were co-transfected to the CRISPRa and CRISPRi in chicken cell lines. The quantitative PCR analysis revealed that the *HMGA1* expression was significantly increased in the VP64 and VPR CRISPRa cell lines when all gRNAs were combined. VP64 displayed the highest level of gene upregulation, with a +22% gene activation. However, there was no significance among the p300, dCas9, dCas9-KRAB, or dCas9-KRAB-MeCP2 cell lines (Figure 2A).

For *SMARCB1*, the three gRNAs were designed at locations within −296 to −85 bp (1, 2, and 3). The quantitative analysis shows that the VP64 and p300 cell lines represented significant upregulation. The VP64 cell line displayed the highest level of gene upregulation with a +62% gene activation; however, there were no significant values observed for the VPR and CRISPRi cell lines (Figure 2B). For *IRF7*, five different locations of gRNAs (1, 2, 3, 4 and 5) were designed spanning from −256 to +163 for gene activation or repression. As results, the expression of *IRF7* was significantly upregulated in the VP64 and VPR CRISPRa cell lines; however, there was no significant gene upregulation in the p300 CRISPRa cell line. On the other hand, in the CRISPRi cell lines, the *IRF7* gene expression in dCas9 (−67%) and dCas9-KRAB (−58%) was significantly downregulated in comparison to mock control; however, there was no significance within the dCas9-KRAB-MeCP2 CRISPRi cell line (Figure 2C). *PPARG* had gRNA locations designed within −230 to −36 bp. *PPARG* was only tested in CRISPRa cell lines due to lower expression and significant upregulation was observed among each of the CRISPRa cell lines. The VP64 cell line displayed the highest level of gene upregulation, with a +65% activation (Figure 2D).

### 3.3. Effects of Individual gRNAs for Gene Regulation

To further validate the effects of the location of each individual gRNAs in CRISPRa and CRISPRi cell lines, each gRNA targeting *SMARCB1* and *IRF7* was separately transfected in CRISPRa and CRISPRi cell lines. Three gRNAs targeting near the TSS of *SMARCB1* gene were examined (Figure 3A), and the results showed that only gRNA 3 induced a significantly higher *SMARCB1* expression in VP64 and p300 CRISPRa cell lines (Figure 3B). On the other hand, gRNA 2 could only downregulate the gene expression in KRAB and MeCP2 CRISPRi cell lines (Figure 3C).

For *IRF7* gene regulation, five individual gRNAs targeting near the TSS of *IRF7* gene were examined (Figure 4A). Based on the results, only gRNA 2 induced a significantly higher *IRF7* expression in VP64 and VPR CRISPRa cell lines, while it was not significant in the p300 CRISPRa cell line. In VP64 and VPR, gRNA 2-transfected samples were significantly increased by 428% and 518% compared to mock controls, respectively (Figure 4B). The same five individual gRNAs were examined in CRISPRi cell lines, and significant *IRF7* downregulation among gRNAs 1, 4, and 5 in dCas9 was observed, with the most downregulation being observed in gRNA 5-transfected samples with a −63% reduction. In the dCas9-KRAB cell line, all of the individual gRNAs induced a significant *IRF7* downregulation and the most downregulation was observed in gRNA 4-transfected samples with a −71% reduction. However, no significant downregulation was observed in the dCas9-KRAB-MeCP2 cell line (Figure 4C).

### 3.4. Transcriptomic Profile of IRF7-Regulated DF-1 Cells

To explore the global effect of the CRISPRa and CRISPRi systems in chicken cells, the transcriptomic profiles of the *IRF7*-regulated DF-1 cells were analyzed by RNA sequencing and subsequent informatic analysis. The dCas9-VPR and dCas9-KRAB were selected based on their expression modulation efficacy and all five gRNAs were introduced to activate (32% upregulation) or repress (58% downregulation) the *IRF7* expression in the cell lines. In CRISPRa cell lines, 259 genes were differentially expressed—200 were upregulated and 59 were downregulated (FDR < 5%) (Figure 5A). In CRISPRi cell lines, a total of 644 genes were DEGs, with 159 upregulated and 485 downregulated as a result of the targeted downregulation of *IRF7* (Figure 5B). One potential off-target transcript was included in the DEG listed, vinexin-like *(LOC107050638),* in dCas9- KRAB cell line (FDR < 5%) (Appendix A).

Functional enrichment analysis of the DEGs was conducted by DAVID to identify the effect of the targeted *IRF7* gene modulation. The effect of *IRF7* upregulation was seen in the enriched GO:biological process (BP) terms in the genes from the CRISPRa cell lines. These terms were “Extracellular matrix organization”, “Cell adhesion”, and “Collagen fibril organization”. In the KEGG pathway, “ECM-receptor interaction”, “Fatty acid metabolism”, and “PPAR signaling pathway” were considerably enriched (Figure 5C). Knockdown of *IRF7* showed the function of *IRF7*, the enriched in GO:BP terms of “Defense response to virus”, “Translation”, and “Immune system process” were found in the genes from the CRISPRi cell lines. Significant enrichment was found in the KEGG pathways known as “Influenza A”, “Herpes simplex virus 1 infection”, and “Ribosome” (Figure 5D).

## 4. Discussion

Targeted gene regulation is an efficient method for investigating the function of the genome in relation to important phenotypes in living organisms [47]. The importance of tools that allow for functional studies such as this will allow for greater utilization in farm animals, which are a valuable protein resource in human society [48]. Particularly, poultry has substantial benefits in academic fields as well as industrial areas due to its unique developmental and reproductive characteristics [49]. However, there are still limitations to identifying specific genetic functions by gene editing, such as changes in genetic information [24]. Therefore, we applied diverse dCas9-mediated CRISPRa and CRISPRi systems to chicken cells and compared for further practical uses in this study. An advantage of the CRISPRa system is the low risk of off-target effects as well as the upregulation of the endogenous genes in their native context [1]. CRISPRi enables the downregulation of multiple genes independently and it circumvents the potential adverse effect of CRISPR knockout [16,50]. An additional advantageous feature of both CRISPRa and CRISPRi is the reversibility of gene manipulation.

To evaluate the effective use of the CRISPRa and CRISPRi systems in chickens, DF-1 cell lines expressing the CRISPRa and CRISPRi components were established. To induce robust expression of the components, the CRISPR/Cas9-NHEJ-mediated gene targeting method and the *piggyBac* transposition-mediated transgenic technique were adopted for CRISPRa and CRISPRi [34,46], respectively, which have been determined as efficient genome-editing methods in chicken [51,52]. To acquire robust exogenous gene expression, the intergenic 3′ region of chicken—the housekeeping gene *GAPDH*—was targeted for insertion of the CRISPRa components. Through NHEJ and targeting *GAPDH,* the integration of CRISPRa components into the targeted regions was maintained in the established cell lines. The *piggyBac* transposition-mediated CRISPRi integration was also successfully achieved, suggesting the versatility of the gene insertion methods in chicken DF-1 cells.

Combination gRNAs were introduced to validate the CRISPRa and CRISPRi DF-1 cell lines for endogenous chicken gene regulation. The results displayed that the introduction of gRNAs targeting the TSS significantly upregulates gene expression. On the contrary, the introduction of gRNAs targeting *HMGA1* and *SMARCB1* could not downregulate gene expression. Relatively higher expression levels of *HMGA1* and *SMARCB1* were observed throughout the data and could be the reason why there was not significant downregulation. Both significant upregulation and downregulation were observed in *IRF7.* Based on these results, the CRISPRa and CRISPRi systems could be applicable in chicken DF-1 cells and the targeting regions of gRNAs are critical for gene regulation.

This research reveals that each effector shows a variable range of gene regulation. VPR allows for strong multi-genome activation, indicating expression levels several times higher than VP64 in the gene panel, while p300 did not show improved levels of activation compared to other activators [53]. In another human cell line, human embryonic kidney 293T (HEK 293T), VPR showed 22- to 320-fold improved activation compared to the VP64 activator [13]. In this research, VP64 and VPR were more effective than p300, although there is little significant difference between VP64 and VPR. p300 involves the use of a histone acetyltransferase (HAT). HAT is responsible for catalyzing the acetylation of histones, which are involved in the regulation of gene expression and the difference in gene activation mechanism may be the reason for discrepancy between the effectors [13,54,55].

In terms of CRISPRi, *SMARCB1* displayed significant downregulation in gRNA2 in the dCas9-KRAB. CRISPRi gRNAs were exclusively efficient in promotor regions near the TSS, reducing the possibility of off-target effects from transcriptional interference elsewhere in the genome [56]. A few studies show that CRISPRi gRNAs are effective near TSS [16,56], but this limits the effectiveness of transcript production for genes that have TSSs that are not well characterized or genes that have many TSSs. *IRF7* had significant downregulation in the dCas9 cell line for gRNAs 1, 4, and 5, and the dCas9-KRAB cell line had the most significance among all of the gRNAs. It has been determined that the optimal inhibition can be induced with gRNAs designed within a window of −50 to +300 bp [16]. Therefore, the results of *IRF7* in the dCas9 cell line also support that the gRNA location is critical and the gene regulation effect is independent of the DNA strand targeted [21]. Nevertheless, it was found that gRNAs that did not fall into the window could also downregulate the gene expression (gRNA1 in dCas9 and gRNA2 in dCas9-KRAB). This suggests that gRNAs targeting not only the optimal window but other parts of the gene, including exons and introns, could also induce the downregulation of the gene expression. Other studies have found that by targeting an intron, there has been less opportunity for off-target effects, along with having a three-fold increase in knock-in efficiency [23,57]. Therefore, the results suggest that the differential gene-regulating effects of the *IRF7* gRNA2 and gRNA3 are caused by effector proteins.

The fusion of dCas9-KRAB was highly efficient, showing a significant downregulation amongst each of the gRNAs and the combination. The mechanism used by dCas9-KRAB to enhance transcriptional repression is that the KRAB domain interacts with KAP1, which ultimately recruits a variety of corepressors including heterochromatin protein 1 (HP1), histone deacetylases, and SET domain bifurcated histone lysine methyltransferase 1 (SETDB1) [21]. While there are data in humans and mice to demonstrate that the dCas9-KRAB-MeCP2 fusion protein is the most effective, that was not replicated in this study [21]. MeCP2 utilizes a different set of transcriptional repressors than KRAB, which include DNA methyltransferase DNMT1 and the SIN3A-histone deacetylase corepressor complex [58,59,60]. Therefore, the difference in transcriptional machinery of KRAB and MeCP2 may indicate why the dCas9-KRAB-MeCP2 cell line was unable to effectively downregulate the genes in this study. Another potential reason for unsuccessful downregulation could be the limited cross activity of the rat MeCP2 in chicken, since it has a low amino acid sequence identity (42%) compared to the chicken MeCP2 [21].

To investigate the effects of individual gRNAs on gene regulation, *SMARCB1* and *IRF7* were further examined in CRISPRa and CRISPRi cell lines. Three gRNAs were designed for *SMARCB1* and five gRNAs were designed for *IRF7* in different locations, either on the positive or negative strand, to determine if different locations or directions were more efficient. For the CRISPRa system, only gRNA3 significantly increased *SMARCB1* expression in the VP64 and P300 cell lines and for *IRF7*, gRNA2 increased gene expression significantly in the VP64 and VPR cell lines. Both of these gRNAs were located close to the TATA box. The result indicates that the location of gRNA is critical for gene regulation in dCas9-mediated gene regulation platforms, which corresponds to the previous research describing optimal gRNA location for CRISPRa (from −400 to −50 bp upstream of the TSS) [53].

Collectively, these results indicate that the diverse effector domains for the dCas9 toolkit are applicable in the chicken system with a variable range of gene regulation. Further studies using additional effector domains that were not validated in this study, such as SunTag, SAM, LSD1 or DNMT1 and a combination of gRNAs, could help optimize chicken gene regulation.

To test the potential use of the CRISPRa and CRISPRi systems as a platform for functional studies in chickens, activation and repression of chicken *IRF7* were further tested to compare with our previous studies [61,62,63]. Through RNAseq, transcriptomic profiles of *IRF7*-regulated DF-1 cells were further validated. *IRF7* is known as the master transcription factor of the type I interferon response in mammals and birds, and *IRF7* gene knockout resulted in increased virus replication with significant downregulation of antiviral activities [63].

In this research, the downregulation of *IRF7* by CRISPRi resulted in the downregulation of interferon stimulated genes that are crucial in antiviral activity such as 2′-5′-oligoadenylate synthetase-like (*OASL*), Toll-like receptor 3 (*TLR3*), interferon induced with helicase C domain 1 (*IFIH1*, as known as *MDA5*), interferon-induced protein with tetratricopeptide repeats 5 (*IFIT5*), myxovirus (influenza virus) resistance 1, interferon-inducible protein p78 (*MX1*), and signal transducer and activator of transcription 1 (*STAT1*). On the other hand, the *IRF7* upregulation by CRISPRa stimulated the genes relating to cell–cell interaction such as integrin subunit α 8 (*ITGA8*), cadherin 11 (*CDH11*) and collagen type VI α 2 chain (*COL6A2*), suggesting the genes’ roles in viral defense by modifying cell surface moiety and also corresponding to previous research [61,62,63].

In summary, the CRISPRa and CRISPRi systems using diverse effectors in chicken DF-1 fibroblast cells were successfully evaluated in this study, and significant results were observed when evaluating the combination of gRNAs and individual gRNAs in both the CRISPRa and CRISPRi cell lines. With the combination of bioinformatic analysis, the parallel studies of CRISPRa and CRISPRi using diverse effector domains are expected to not only contribute to regulating specific gene expression but also help to annotate the functional elements, including enhancer and repressor regions in the chicken genome. Further studies are warranted to utilize different cell lines to address the different biological contexts of gene expression.

## Figures and Tables

**Figure 1 genes-14-00906-f001:**
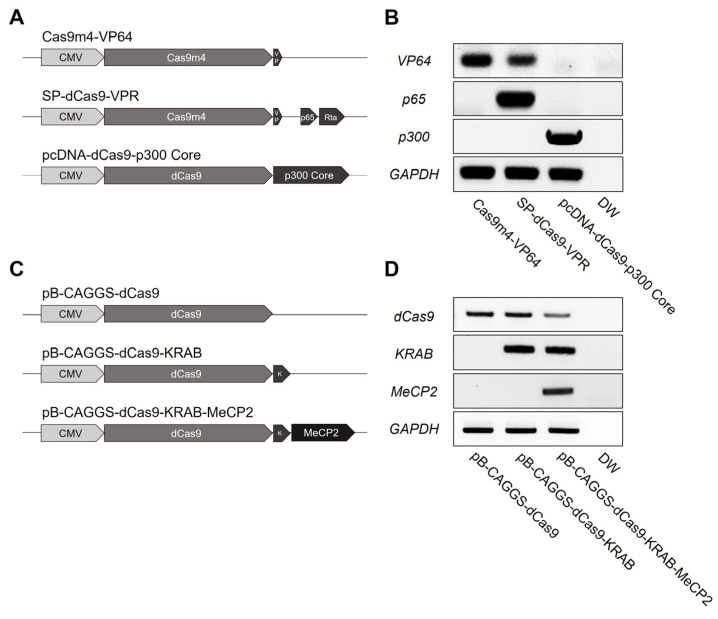
Establishment of CRISPRa and CRISPRi expressing cell lines (**A**) Schematic representation of CRISPRa vectors. CMV, cytomegalovirus promoter; Cas9m4, inactive Cas9 lacking endonuclease activity; VP, activation domain VP64; Rta, Epstein–Barr virus R transactivator; p300 Core, p300 histone acetyltransferase (**B**) RT-PCR analysis in the established CRISPRa DF-1 cell lines. (**C**) Schematic representation of CRISPRi vectors. dCas9, inactive Cas9 lacking endonuclease activity; K, Krüppel-associated box, KRAB; MeCP2, methyl-CpG binding protein 2 (**D**) RT-PCR analysis in the established CRISPRi DF-1 cell lines. The endogenous *GAPDH* gene was used as a control.

**Figure 2 genes-14-00906-f002:**
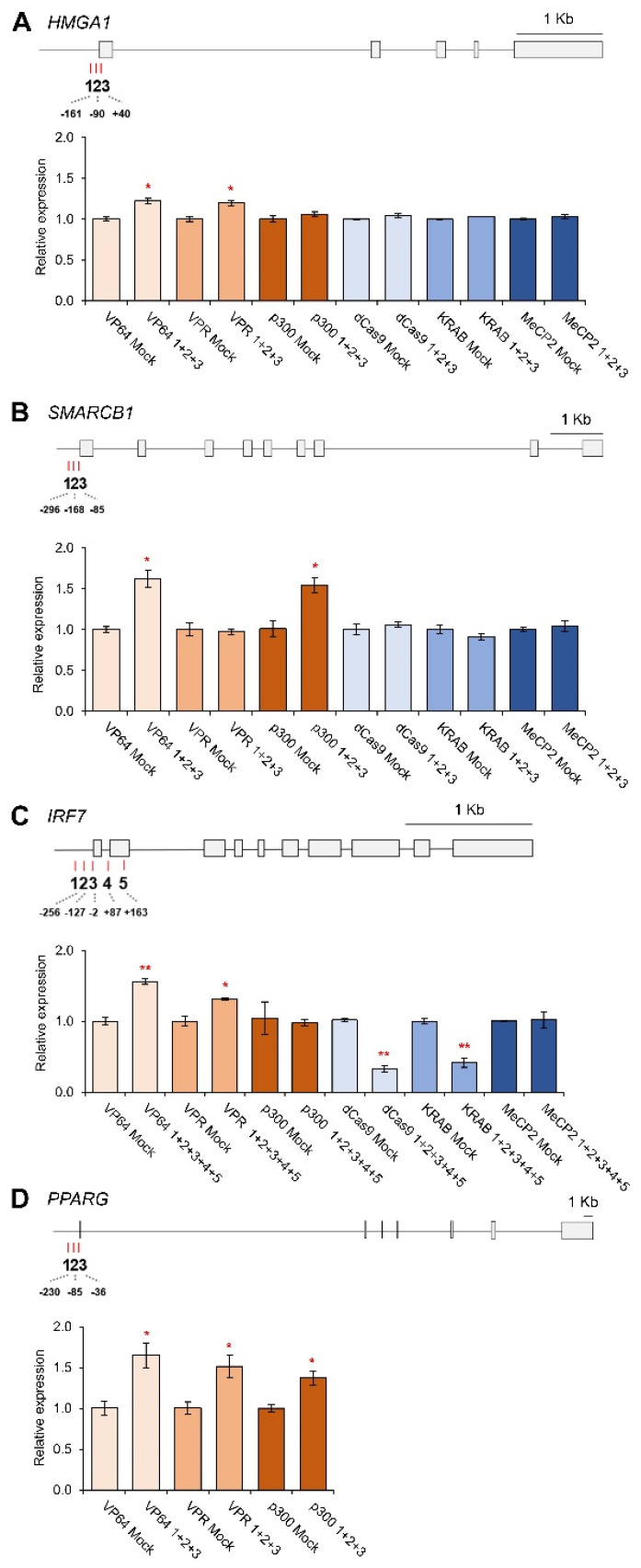
The regulation of the chicken *HMGA1, SMARCB1, IRF7* and *PPARG* gene expression in CRISPRa and CRISPRi DF-1 cells. Gene structure and quantitative analysis of *HMGA1* (**A**), *SMARCB1* (**B**), *IRF7* (**C**) and *PPARG* (**D**) in the gRNAs-treated CRISPRa and CRISPRi cell lines. The targeting positions of the gRNAs for each gene (1, 2 and 3 or 1, 2, 3, 4 and 5) were marked. Grey rectangles indicate exons. Scale bars, 1 Kb. Data are shown as the mean ± SEM. Significant differences between the mock controls and gRNAs-transfected samples were determined by Student’s *t*-test. Statistical significance was marked as * *p* < 0.05, ** *p* < 0.01.

**Figure 3 genes-14-00906-f003:**
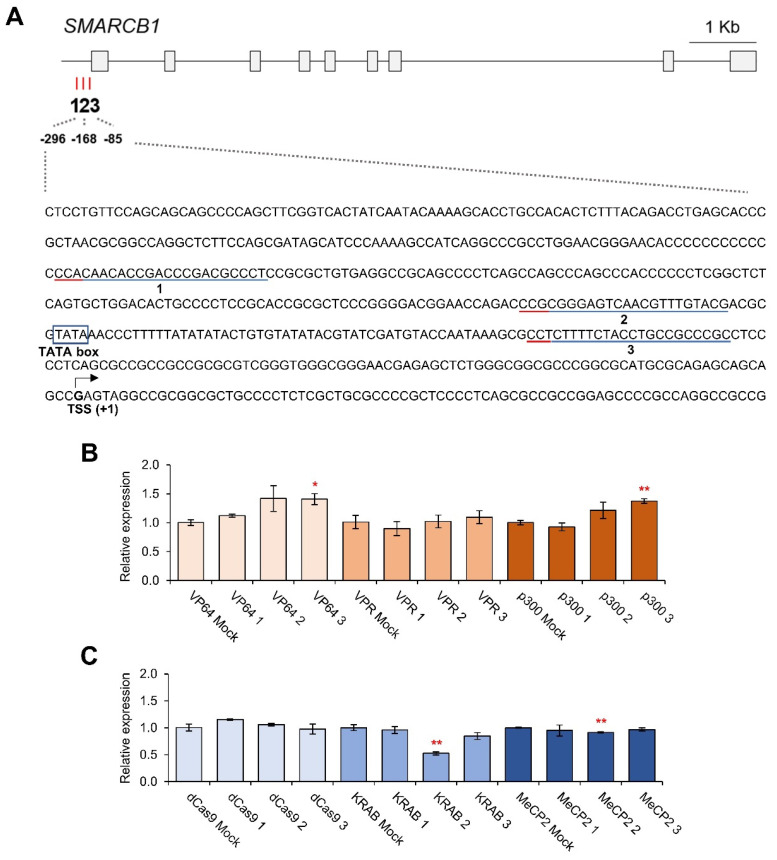
The regulation of the chicken *SMARCB1* gene expression by individual gRNAs in CRISPRa and CRISPRi DF-1 cells. (**A**) Gene structure of *SMARCB1* gene and the location of gRNAs (1, 2 and 3). The number indicates the position of gRNAs. Grey rectangles indicate exons. Scale bars, 1 Kb. TATA box and transcription start site (TSS) are marked. The blue bar indicates the gRNA binding site, and the red bar indicates the protospacer adjacent motif sequence (PAM). Quantitative analysis in the individual gRNA-treated CRISPRa DF-1 cell lines (**B**), and CRISPRi DF-1 cell lines (**C**). Data are shown as the mean ± SEM. Significant differences between the mock controls and gRNAs-transfected samples were determined by Student’s *t*-test. Statistical significance was ranked as * *p* < 0.05, ** *p* < 0.01.

**Figure 4 genes-14-00906-f004:**
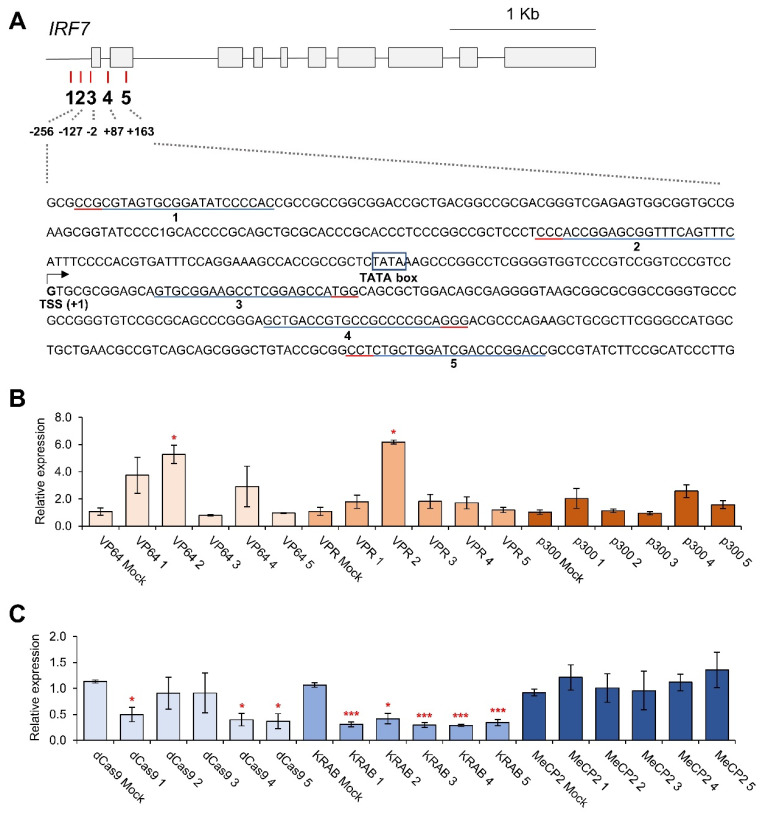
The regulation of the chicken *IRF7* gene expression by individual gRNAs in CRISPRa and CRISPRi DF-1 cells. (**A**) Gene structure of *IRF7* gene and the location of gRNAs (1, 2, 3, 4 and 5). The number indicates the position of gRNAs. Grey rectangles indicate exons. Scale bars, 1 Kb. TATA box and transcription start site (TSS) are marked. The blue bar indicates the gRNA binding site, and the red bar indicates the protospacer adjacent motif sequence (PAM). Quantitative analysis in the individual gRNA-treated CRISPRa DF-1 cell lines (**B**), and CRISPRi DF-1 cell lines (**C**). Data are shown as the mean ± SEM. Significant differences between the mock controls and gRNAs-transfected samples were determined by Student’s *t*-test. Statistical significance was ranked as * *p* < 0.05, *** *p* < 0.001.

**Figure 5 genes-14-00906-f005:**
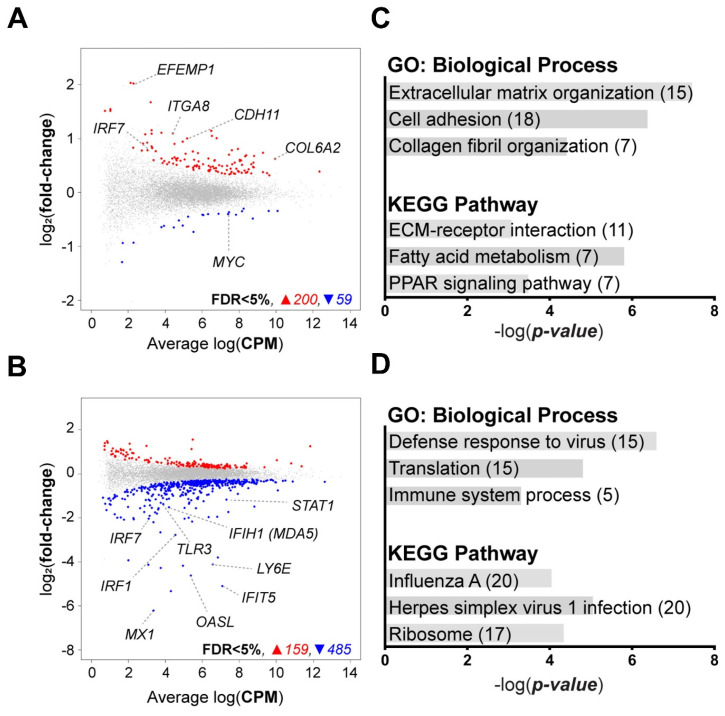
Transcriptomic profiles in the *IRF7*-regulated DF-1 cell lines. Volcano plots of the transcriptomic differences between *IRF7*-upregulated DF-1 cells (VPR) and mock controls (**A**), and between *IRF7*-downregulated DF-1 cells (KRAB) and mock controls (**B**). The colored dots correspond to significant differentially expressed genes (DEGs; False discovery rate (FDR) < 5%). The numbers of DEGs are shown in the bottom right corner of each plot. Gene ontology (GO) and pathway analysis by DAVID using DEGs from the *IRF7*-upregulated (**C**) or -downregulated DF-1 cells (**D**). The number of genes enriched in each biological process is in parentheses.

## Data Availability

RNASeq data are available through, GEO Accession number GSE217310.

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
