# Peer review of "Targeted Modulation of Chicken Genes In Vitro Using CRISPRa and CRISPRi Toolkit"

_genes, 2023, doi:10.3390/genes14040906_

Round 1
Reviewer 1 Report
It is interesting to note that the authors of this manuscript have focused on evaluating the performance of CRISPRa and CRISPRi technology in a chicken cell line, which is an area of research where the use of these techniques has been limited. By evaluating a selected list of effector domains and gRNA target sites, the authors were able to identify variations in activity levels, which could provide valuable information for researchers interested in using these techniques in avian studies. The targeted feature and irreversibility of CRISPR technology make it a powerful tool in genetic research, and this study contributes to our understanding of how this technology can be used effectively in avian research. Overall, this study has the potential to advance the field of avian genetics and inform future research in this area. In the meantime, I also have some major and minor concerns listed below:
· In figure 2A, dCas9+gRNA mix showed significant repression but dCas9-MeCP2 +gRNA mix did not. Although similar results was observed in figure 4C, it is confusing why this difference happened. Is it possible that the constructs had some issue expressing these gRNAs? The authors should comment on this difference.
· In figure 3B, gRNA 2 showed about 5fold increase in IRF7 expression. gRNA 1 and gRNA 4 also showed significantly 2-4 fold increase with larger variation. However, in the mixed experiment presented in figure 2C, the increase in IRF7 expression in the corresponding CRISPRa groups is not as significant. The authors please comment on this difference. Also please comment on why using the gRNA mix treated sample for RNAseq rather than using the more potent IRF7 gRNA 2 only?
· QC plots of the bulk RNAseq replicates should be included in the supplements to show the quality of sequencing data. This is critical in determining the DEGs identified in the manuscript, especially when the authors only used 2 replicates for group.
Minor:
· line 54: CRIPSRi (dCas9, KRAB, and KRAB-MeCP2). it is confusing why dCas9 is included in the parenthesis.
· Line 335-336, HEK 293T is the same celline that was discussed in line 330-line 334.
· In section 3.2, the author did not report the examination window of gene expression post gRNA delivery. The examination timepoint is critical to making conclusions. The authors should include this detail.
Author Response
On behalf of the authors, I would like to thank you for your review of our manuscript entitled “Targeted modulation of chicken genes in vitro using CRISPRa and CRISPRi toolkit”. We would also like to thank you for the constructive and valuable comments, and hope that you will find the completion of the recent manuscript changes is satisfactory and acceptable for publication in Genes. Each comment has been responded point by point in the following. Updated manuscript file is also attached.
- In figure 2A, dCas9+gRNA mix showed significant repression but dCas9-MeCP2 +gRNA mix did not. Although similar results was observed in figure 4C, it is confusing why this difference happened. Is it possible that the constructs had some issue expressing these gRNAs? The authors should comment on this difference.
The dCas9 and gRNA mix was shown in figure 2C, which correlates with the data shown in figure 4C. Ultimately, it was expected that MeCP2 would be the most effective at gene regulation, but the difference was commented on in lines 386-394.
- In figure 3B, gRNA 2 showed about 5 fold increase in IRF7 expression. gRNA 1 and gRNA 4 also showed significantly 2-4 fold increase with larger variation. However, in the mixed experiment presented in figure 2C, the increase in IRF7 expression in the corresponding CRISPRa groups is not as significant. The authors please comment on this difference. Also please comment on why using the gRNA mix treated sample for RNAseq rather than using the more potent IRF7 gRNA 2 only?
IRF7 expression was presented in figure 4, with individual gRNA 2 showing a significant increase in figure 4B. We have now outlined the specific amounts of each gRNA in lines 146-149. For IRF7 combination gRNAs, 2 ug was used as the total – 400 ng per each of the 5 gRNAs. For SMARCB1, 1.5 ug was used total – 500 ng per each of the 3 gRNAs. The potential reason for the variation when evaluating individual gRNA vs. combination gRNA could be due to the concentrations. Essentially, when using combination gRNA, each gRNA was diluted in comparison to when using just the individual gRNA.
The reason that gRNA mix was utilized for RNAseq is outlined in lines 282-285.
- QC plots of the bulk RNAseq replicates should be included in the supplements to show the quality of sequencing data. This is critical in determining the DEGs identified in the manuscript, especially when the authors only used 2 replicates for group
Figure S2, PCA plots are added to show the bulk RNAseq QC.
Minor:
- line 54 (68?): CRIPSRi (dCas9, KRAB, and KRAB-MeCP2). it is confusing why dCas9 is included in the parenthesis. Line 97-98
There was 3 total vectors constructed for CRISPRi – a piggyBac transposon vector containing dCas9, a PB vector containing dCas9-KRAB, and a pB vector containing dCas9-KRAB-MeCP2 (outlined in lines 104-105). We have adjusted the parenthesis in line 75-76, and now have the CRISPRi systems identified as dCas9, dCas9-KRAB, and dCas9-KRAB-MeCP2.
- Line 335-336 (350?), HEK 293T is the same celline that was discussed in line 330-line 334.
This was adjusted in lines 355-356.
- In section 3.2, the author did not report the examination window of gene expression post gRNA delivery. The examination timepoint is critical to making conclusions. The authors should include this detail.
RNA was extracted from the cells 48-72 hours post-transfection. This is mentioned in section 2.5, in lines 145-147.

Reviewer 2 Report
Authors report targeted engineering of CRISPT/Cas9 gene editing system into regulating gene expression. Using DF-1 cell line, CRISPRa and CRISPRi cell lines were developed by stable transfection of various molecular effector constructs and gRNAs for several genes were evaluated for controlling gene expression. Lastly, RNA-seq analysis was conducted to find transcriptomics effects of engineered control of gene expression.
The study is in the scientific interest and below include revising suggestions:
- Lines 64 – 67, 324: How this targeted gene expression control method can be used in chicken in vivo? Which topics of research, such as growth, reproduction, safety, or health, can be applied with this technology suggested in this study? What is the benefit of this method compared with dsRNA based-RNA interference approach (particularly down-regulating gene expression)? Authors may need to add more details of the importance and usefulness of this technology (targeted control of gene expression using CRISPR/Cas9 gene editing system) for in vivo approaches in avian/poultry researches.
- Lines 325 – 343, 390 - 397: Those discussions are repeated to result section. They can be shortened.
Author Response
On behalf of the authors, I would like to thank you for your review of our manuscript entitled “Targeted modulation of chicken genes in vitro using CRISPRa and CRISPRi toolkit”. We would also like to thank you for the constructive and valuable comments, and hope that you will find the completion of the recent manuscript changes is satisfactory and acceptable for publication in Genes. Each comment has been responded point by point in the following. Updated manuscript file is also attached.
- Lines 64 – 67, 324: How this targeted gene expression control method can be used in chicken in vivo? Which topics of research, such as growth, reproduction, safety, or health, can be applied with this technology suggested in this study? What is the benefit of this method compared with dsRNA based-RNA interference approach (particularly down-regulating gene expression)? Authors may need to add more details of the importance and usefulness of this technology (targeted control of gene expression using CRISPR/Cas9 gene editing system) for in vivo approaches in avian/poultry researches.
A paragraph has been added to the introduction outlining how this technology can be utilized in chickens. This technology is very versatile and can be utilized for productivity, disease resistance, and can also have an influence on animal welfare concerns.
The benefits of this CRISPR method are the ability for targeted gene regulation. The individual benefits of the CRISPRa/CRISPRi systems are identified in lines 318-319 and 326-331.
- Lines 325 – 343, 390 – 397: Those discussions are repeated to result section. They can be shortened.
The information presented in these sections do have differences – in lines 332-351, we are stating that the integration of these vectors into CRISPRa and CRISPRi cells was successful. We did clarify in line 343 that this section is representing the discussion of combination gRNAs. We then discuss the trends observed from the combination gRNAs.
The information presented in lines 395-405 identify the effects of individual gRNAs. SMARCB1 and IRF7 were both transfected with individual gRNAs, and the outcome of those transfections were commented on.
